# Spontaneous Splenic Rupture Secondary to Infectious Mononucleosis

**DOI:** 10.3390/diagnostics14141536

**Published:** 2024-07-16

**Authors:** Ismini Kountouri, Evangelos N. Vitkos, Periklis Dimasis, Miltiadis Chandolias, Maria Martha Galani Manolakou, Nikolaos Gkiatas, Dimitra Manolakaki

**Affiliations:** Department of General Surgery, General Hospital of Katerini, 60100 Katerini, Greece; envitkos@gmail.com (E.N.V.); dimasis@yahoo.com (P.D.); miltoshandolias@gmail.com (M.C.); mariamarthagalani@gmail.com (M.M.G.M.); nikgiat71@gmail.com (N.G.); dimanolakaki@gmail.com (D.M.)

**Keywords:** splenic rupture, infectious mononucleosis, Epstein–Barr virus, hemoperitoneum

## Abstract

Spontaneous splenic rupture (SSR) is a relatively rare but potentially lethal complication of infectious mononucleosis (IM). While SSR is extremely rare in patients with proven IM, it is the most lethal complication of the infection (9% mortality rate) and can present completely asymptomatically or with abdominal pain and hemodynamic instability. As adolescents and young adults are the most affected population group, with this case report, we intend to raise the vigilance of any doctor treating those patients in the emergency department. We present the case of a 16-year-old patient with an atraumatic splenic rupture and hemoperitoneum secondary to an Epstein–Barr virus (EBV) infection. The patient underwent an exploratory laparotomy, and a splenectomy was performed. This case demonstrates that, even if SSR in patients with IM is extremely rare, it should always be considered in a patient with a relevant clinical presentation.

We present the case of an SSR secondary to an EBV infection in a 16-year-old male patient. He presented in the emergency department complaining of acute upper left quadrant abdominal pain without any other reporting symptoms. The patient had been diagnosed three weeks prior to admission with IM with a positive monotest and was recovering at home ever since. Upon the IM diagnosis, the patient reported a sore throat, was mildly febrile, and had cervical lymphadenopathy. No splenomegaly was detected upon the first presentation, while his laboratory tests found his hemoglobin (Hb) at 14.1 g/dL, his hematocrit (HCT) at 42.3%, and the white blood cell count (WBC) at 14.1 × 10^3^/μL, with 33.1% neutrophils and 50.8% lymphocytes. His biochemical profile showed an increase in the values of serum glutamic pyruvic transaminase (SGPT) at 181.0 U/L and serum glutamic oxalacetic transaminase (SGOT) at 90.0 U/L. The patient had no significant medical history and was taking no specific medication. No history of trauma was reported. When presented in the ER, three weeks later, with abdominal pain, he was mildly tachycardic (90 bps/min), with evidence of localized peritonism on his left upper quadrant. Regarding IM, no other symptoms were described. The first laboratory tests upon admission found his hemoglobin (Hb) at 13.1 g/dL, his hematocrit (HCT) at 39.1%, and the white blood cell count (WBC) slightly elevated at 14.1 × 10^3^/μL with 21.1% neutrophils and 71.5% lymphocytes. His complete blood count (CBC) upon admission is shown in Table 1.

An urgent abdominal ultrasound was performed that revealed an enlargement of the spleen (21 cm in diameter) with a large subcapsular hematoma (13 cm in diameter) and a significant amount of complex fluid within the abdominal cavity. In addition, and with the patient remaining hemodynamically stable, an abdominal Computer Tomography scan was performed and the enlargement of the spleen with a maximum diameter of 21.60 cm and the subcapsular hematoma were confirmed (Figure 1).

While the multidetector CT enables the detection of splenic damage and hemorrhage [1], in hemodynamically unstable patients with a potential splenic rupture, a fast-track ultrasound for the detection of hemoperitoneum is the first approach, although its sensitivity is limited for the detection of the rupture (72–78%) [1,2].

The patient was monitored closely for approximately one hour, and a second blood test found the Hb to be 10.0 g/dL. Because of the drop in the Hb levels, the decision for an exploratory laparotomy was made. An emergency laparotomy was performed and a large amount of hemoperitoneum was evacuated from the abdominal cavity. The spleen was found enlarged, with a subcapsular hematoma and a ruptured capsule. An emergent splenectomy was performed and the hemoperitoneum was evacuated (Figure 2). No other intra-abdominal pathology was noted. The patient received only crystalloid fluids and no blood transfusions. He recovered without any post-operative complications. His hemoglobin settled spontaneously in the following days, and, gradually, he became afebrile. Prior to discharge, he received the Pneumococcal, Meningococcal, and Hemophilus influenzae (Hib) vaccinations and was prescribed an appropriate post-splenectomy antibiotic prophylaxis. A pathologic examination revealed an enlarged spleen, weighing 820 g, with two large subcapsular hematoma (with a maximum diameter of 13 cm) and a large laceration on its surface. The red pulp was enlarged because of an infiltration with a heterogeneous population of small lymphocytes, activated lymphoid cells, immunoblasts, plasma cells, and histiocytes. The diagnosis of the SSR secondary to IM was confirmed.

The management of a splenic injury is generally a debate [1]. It depends on the hemodynamic status, resource availability, splenic injury grade, and presence of comorbidities [1,2]. If possible, splenic preservation is to be preferred, especially in younger patients, to minimize the risk of post-splenectomy infections and septicemia [1,3]. In hemodynamically stable patients with CT findings of active contrast extravasation, endovascular techniques such as splenic artery angiography with embolization to enable the preservation of the spleen may be the treatment of choice [2,4]. A splenic rupture requiring late surgery has been reported in patients whose initial CT shows a low-grade splenic lesion [1,5], so, if all the conservative treatments fail and the patient becomes hemodynamically unstable, a surgical exploration is recommended [2]. When a surgical approach is necessary, the preservation of the spleen is to be pursued, with splenorraphy being the suggested technique [1,2,3]. Splenectomy and the re-implantation of the spleen are also reported in the literature as a method of spleen preservation in cases of splenic damage, but the procedures should be performed by experienced surgeons [1,2,3].

This case demonstrates the risk of an atraumatic splenic rupture in infectious mononucleosis and shows the importance of prompt diagnosis and appropriate counselling in patients with IM. IM is usually a self-limiting illness, but its clinical presentation may be variable, and the classical triad may be absent [6]. The most significant finding associated with IM is hepatosplenomegaly [7]. An SSR complicating IM is a rare complication, occurring in 0.1–0.5 percent of patients with proven IM [8]. During the EBV infection, the mononuclear cells collect within the lymphoid tissue, causing the enlargement of the spleen [2]. In about 50% of patients with splenomegaly, as the spleen enlarges, the splenic capsule thins and the spleen is vulnerable to laceration or rupture [2]. The preservation of the splenic tissue is to be pursued to avoid post-splenectomy infections and sepsis [2,3].

Even with all the limitations, our case report showcases an extremely rare but potentially fatal complication of IM. Based on the current National Institute for Health and Care Excellence (NICE) guidance, a patient suffering from IM is advised to avoid contact or collision sports or heavy lifting for the first month of the illness (to reduce the risk of splenic rupture) [9]. There is no formal international consensus on when the patients should return to their normal activities, but, since a splenic injury has been reported up to 8 weeks following infection, we advise patients with IM to be cautious for this time period. Healthcare providers should remain vigilant, and a possible splenic rupture should be considered in any abdominal pain in IM.

## Figures and Tables

**Figure 1 diagnostics-14-01536-f001:**
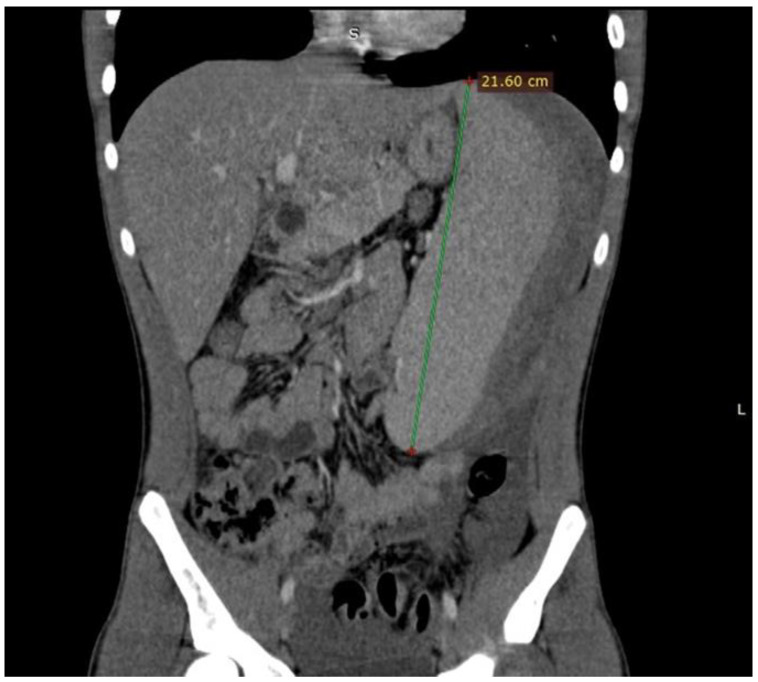
Computer Tomography of the patient showing the enlargement of the spleen with maximum diameter of 21.60 cm and the large subcapsular hematoma.

**Figure 2 diagnostics-14-01536-f002:**
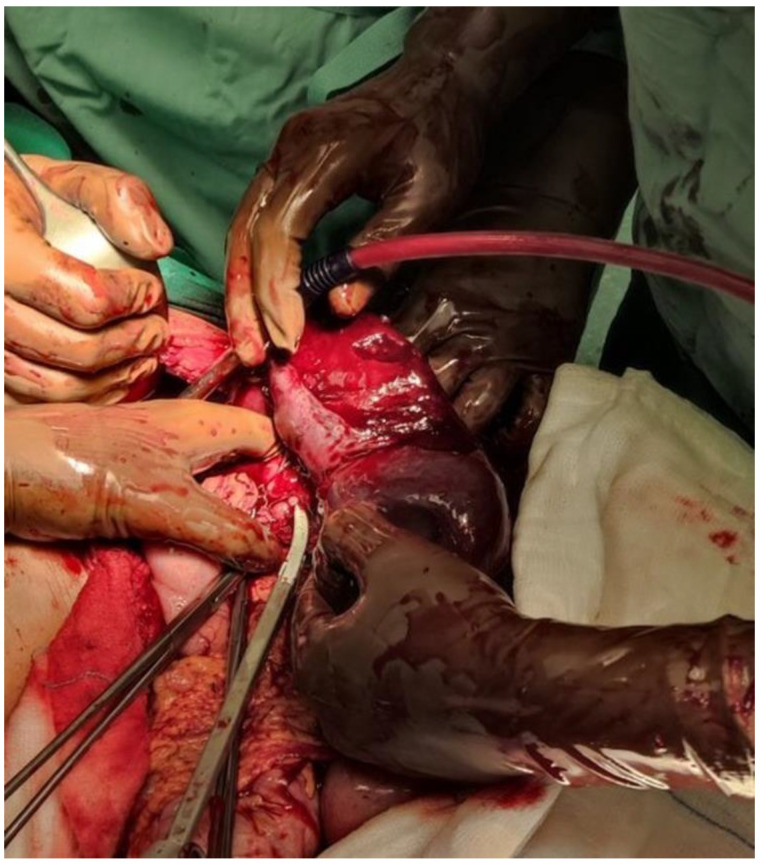
An intraoperative image showing the enlargement of the spleen, the multiple lacerations on the splenic parenchyma, and the rupture of the splenic capsule.

**Table 1 diagnostics-14-01536-t001:** Complete blood count upon admission.

Complete Blood Count	
Red Blood cell (RBC) count	4.61 × 10^6^/μL
White blood cell (WBC) count	14.1 × 10^3^/μL
Hemoglobin (Hb)	3.1 g/dL
Hematocrit (HCT)	39.1%
Patelet (PLT) count	169 × 10^3^/μL

## Data Availability

All relevant data are within the manuscript.

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
