# Peer review of "Spontaneous Splenic Rupture Secondary to Infectious Mononucleosis"

_diagnostics, 2024, doi:10.3390/diagnostics14141536_

Round 1
Reviewer 1 Report
Comments and Suggestions for Authors
1Separate the figure captions from the text.
2Line 39-41. “CT allows the detection and safe characterization of spontaneous splenic lesions, together with the identification or exclusion of active hemorrhage, perisplenic hemorrhage, or hemoperitoneum” Revise this sentece, it sounds redundant.
3The fast-track ultrasound is not a test. Maybe better the first approach….
4Line 73-74. “When a surgical approach is necessary, a preservation of the spleen withsplenorraphy or splenectomy and re-implantation.” Revise this sentence. It sounds strange to me. Spleen re-implantation?? It you are sure about it, please provide any reference.
5Line 82-83. “rupture [5]. of splenic tissue is to be pursued to avoid post splenectomy infections and sepsis [5, 13].” Something is missing in this senteces, revise please.
6Line 85-86. “There is no international formal consensus on when patients should return to normal activities, but a splenic injury has been reported up to 8 weeks post-infection.” Revise this sentece, please.
Author Response
Comment 1 Separate the figure captions from the text.
Response 1 We have separated the captions from the text.
Comment 2 Line 39-41. “CT allows the detection and safe characterization of spontaneous splenic lesions, together with the identification or exclusion of active hemorrhage, perisplenic hemorrhage, or hemoperitoneum” Revise this sentece, it sounds redundant.
Response 2 We revised the text
Comment 3 fast-track ultrasound is not a test. Maybe better the first approach….
Response 3 We revised the text
Comment 4 Line 73-74. “When a surgical approach is necessary, a preservation of the spleen with splenorraphy or splenectomy and re-implantation.” Revise this sentence. It sounds strange to me. Spleen re-implantation?? It you are sure about it, please provide any reference.
Response 4 We revised the text and have added the references regarding the spleen re-implantantion. Please find the references at:
a) Sergent SR, Johnson SM, Ashurst J, Johnston G. Epstein-barr virus-associated atraumatic spleen laceration presenting with neck and shoulder pain. Am J Case Rep. 2015; 16:774–777.
b)Gómez-Ramos JJ, Marín-Medina A, Lisjuan-Bracamontes J, Garciá-Ramírez D, Gust-Parra H, Ascencio-Rodríguez MG. Adolescent with Spontaneous Splenic Rupture as a Cause of Hemoperitoneum in the Emergency Department: Case Report and Literature Review. Pediatr Emerg Care. 2020;36(12): E737–E741.
c)Purkiss SF. Splenic rupture and infectious mononucleosis-splenectomy, splenorrhaphy or non-operative management? J R Soc Med. 1992;85(8):458–459.
Comment 5 Line 82-83. “rupture [5]. of splenic tissue is to be pursued to avoid post splenectomy infections and sepsis [5, 13].” Something is missing in this senteces, revise please.
Response 5 We revised the sentence
Comment 6 Line 85-86. “There is no international formal consensus on when patients should return to normal activities, but a splenic injury has been reported up to 8 weeks post-infection.” Revise this sentece, please.
Response 6 We revised the sentence.
Reviewer 2 Report
Comments and Suggestions for Authors
In the submitted manuscript the authors report a case of spontaneous splenic rupture in a 16-year-old male patient diagnosed with infectious mononucleosis (IM) three weeks before admission in the Emergency Department with localised peritonism. Due to the rarity of this complication the report is very useful from the clinical perspective.
There are several suggestions to be considered:
- A brief description of the clinical picture (disease onset, was the splenomegaly detected at first presentation?) and the laboratory findings at IM diagnosis (include also the results of CBC test among other routine chemistry test)
- Mention the results for all CBC parameters at admission in the Emergency Department
- Please separate the description of Figure 2 from the main text
- If available, include an image from the histopathological exam of spleen
- In the Discussion section please complete the sentence on line 82 “of splenic tissue is to be pursued to avoid post splenectomy infections and sepsis [5, 13].
Author Response
Thank you for your review and comments
Comment 1 A brief description of the clinical picture (disease onset, was the splenomegaly detected at first presentation?) and the laboratory findings at IM diagnosis (include also the results of CBC test among other routine chemistry test)
Response 1 We agree with this comment and have added the information.
Comment 2 Mention the results for all CBC parameters at admission in the Emergency Department
Response 2 We agree with this comment and hab added the information
Comment 3 Please separate the description of Figure 2 from the main text
Response 3 We have revised the text
Comment 4 If available, include an image from the histopathological exam of spleen
Response 4 Unfortunately we have no image of the histopathological exam.
Comment 5 In the Discussion section please complete the sentence on line 82 “of splenic tissue is to be pursued to avoid post splenectomy infections and sepsis [5, 13].
Response 5 We corrected the text.
Round 2
Reviewer 2 Report
Comments and Suggestions for Authors
The authors have complied with the reviewer's requests and the manuscript is worth publishing.